# Neutrophil-To-Lymphocyte and Platelet-To-Lymphocyte Ratios as Prognostic Markers of Survival in Patients with Head and Neck Tumours—Results of a Retrospective Multicentric Study

**DOI:** 10.3390/ijerph17051742

**Published:** 2020-03-07

**Authors:** Zsuzsanna Szilasi, Valéria Jósa, Zsombor Zrubka, Tünde Mezei, Tamás Vass, Keresztély Merkel, Frigyes Helfferich, Zsolt Baranyai

**Affiliations:** 1Department of Otorhinolaryngology and Head and Neck Surgery, HDF Medical Centre, H-1134 Budapest, Hungary; helfferich@yahoo.com; 2Department of Otorhinolaryngology and Head and Neck Surgery, Jahn Ferenc Hospital, H-1204 Budapest, Hungary; josavaleria@gmail.com; 3Department of Health Economics, Corvinus University of Budapest, H-1093 Budapest, Hungary; zsombor.zrubka@uni-corvinus.hu; 4Department of Urology, Jahn Ferenc Hospital, H-1204 Budapest, Hungary; stiffnesskft@gmail.com; 5Department of Surgery, Szent Imre Hospital, H-1115 Budapest, Hungary; dr.vasst@gmail.com (T.V.); kmerkel1@gmail.com (K.M.); 61st Department of Surgery, Semmelweis University, H-1082 Budapest, Hungary; barazso@gmail.com

**Keywords:** head and neck neoplasms, survival, neutrophil-to-lymphocyte ratio, platelet-to-lymphocyte ratio

## Abstract

Background: The neutrophil-to-lymphocyte ratio (NLR) and the platelet-to-lymphocyte ratio (PLR) may be useful for drawing conclusions about the survival of head and neck squamous cell carcinoma (HNSCC) patients. Methods: Clinical data of 156 patients managed for HNSCC at two head and neck surgery centres were analyzed retrospectively. We studied the relationships between survival and PLR as well as NLR. Results: With regards to 5-year survival, the difference between the two groups with PLR values lower or higher than the threshold was statistically significant (*p* = 0.004), and we found the same for disease-free survival (*p* = 0.05), and tumour-specific mortality (*p* = 0.009). Concerning NLR, the difference in tumour-specific survival was statistically significant (*p* = 0.006). According to the multivariate analysis, NLR values higher than the threshold indicated an enhanced risk for overall as well as for tumour-specific mortality. Conclusion: In HNSCC patients, a high NLR may be considered as an independent risk factor for 5-year overall survival.

## 1. Introduction

Head and neck squamous cell carcinoma (HNSCC) of mucosal origin is the sixth most common malignancy, with an estimated 900,000 new cases and about 350,000 fatalities a year worldwide [1]. The availability of an easy-to-determine prognostic factor could improve the effectiveness of individualized therapy.

Certain formed elements of the blood, as well as their ratios, can be used to establish the diagnosis and to predict prognosis.

Malignant tumours influence the number of platelets and alter their functioning. However, the mechanisms of the relationship between thrombocytosis and the tumour have not yet been fully elucidated and several hypotheses have been proposed [2].

The facilitatory role of chronic inflammation in the development and progression of solid tumours have been described for malignancies of the lung, breast, and gastrointestinal tract, the prostate, and the ovaries [3,4,5,6,7], as well as for HNSCC [8,9].

Neutrophils might, in fact, facilitate tumour growth and the formation of metastases—these cells can inhibit the activities of the lymphocytes and thereby attenuate the antitumour immune response [10]. Neutrophil cells can contribute to carcinogenesis and tumour progression in many ways [11]: (1) These cells can release genotoxic substances, which damage the DNA of epithelial cells [12]. (2) They can promote tumour growth by releasing soluble factors and proteases, such as prostaglandin E2 and neutrophil elastase [13,14]. (3) Neutrophils facilitate metastasis formation by enhancing the capabilities of the tumour cells for migration, invasive spread, and colony formation, as well as by breaking down the extracellular matrix [15]. (4) They contribute to the formation of new blood vessels by producing pro-angiogenic factors, such as matrix metalloproteinase-9, and vascular endothelial growth factor [16]. (5) Neutrophils exert immunosuppressive action by inhibiting effector T-cells along with the proliferation and the functioning of natural killer cells, as well the fact that they enhance tumour growth and metastasis formation through the modulation of macrophage activity [17].

These observations suggest that neutrophil-to-lymphocyte ratio (NLR), and platelet-to-lymphocyte ratio (PLR) determined in the blood of patients with a malignancy might be related to tumour growth [18].

Our study intended to ascertain whether pre-treatment NLR and PLR values determined in HNSCC patients are suitable predictors of survival.

## 2. Materials and Methods

The study was approved by the Scientific and Research Ethics Committee of the Hungarian Medical Research Council on 14 December 2015 (Nr. 8951-3/2015/EKU (0444/15).

We conducted a retrospective analysis on the clinical data of patients managed for primary HNSCC at two head and neck surgical centres (Uzsoki Street Hospital, Budapest and Medical Centre of the Hungarian Defence Forces, Budapest), between 2000 and 2014. The criteria for inclusion comprised the diagnosis of squamous cell carcinoma confirmed by histology, and the availability of platelet, neutrophil, and lymphocyte counts measured at the earliest one month before treatment. The exclusion criteria were as follows: the lack of an R0 resection, the presence of a synchronous tumour, inflammatory conditions (fistula of the neck, pneumonia, wound infection, abscess, cholecystitis, cannula sepsis, endocarditis, urinary tract infection, Crohn’s disease, ulcerative colitis, etc.), thromboembolic events (deep vein thrombosis, pulmonary embolism, myocardial infarction), corticosteroid therapy, and treatment with platelet aggregation inhibitors during the month before blood sampling.

The data available on the factors predisposing to malignancy (smoking, alcohol consumption, infection by human papillomavirus or by the Epstein–Barr virus, etc.) were incomplete and hence, we did not take these into account in our study. Overall survival was defined as the period from the first day of treatment to the day of death from any cause—or to the last date of observation. Disease-free survival was calculated as the time from the start date of treatment and to the date of the detection of recurrence. Missing mortality data were supplemented with information obtained from public health databases maintained by the government.

Based on the above criteria, 393 out of the 549 patients were excluded owing to incomplete laboratory data and hence, 156 patients were included in the study. They were staged according to the 7th edition of the TNM classification of the American Joint Committee on Cancer. Due to the occasional uncertainty of the clinical information available, we had to work with diverse survival data. In particular, overall, disease-specific, and tumour-specific survivals were analysed using data available from 156, 150, and 149 patients.

### Statistical Analysis

The relationships among categorical variables were analysed with the cross-table method and the χ^2^ test. The non-parametric Mann–Whitney, U-, and Kruskal–Wallis tests were used to study the associations between the continuous variables PLR, NLR, and the subgroups of patients. The survival of patients with PLR and NLR values lower or higher than the threshold was analysed with the log-rank test and plotted as a Kaplan–Meier survival curve. The correlations between PLR, NLR, and survival were controlled for age, gender, and the characteristics of the malignancy, and then subjected to multivariate Cox regression analysis. The joint effect of discrete, multiple categorical variables was appraised with the Wald test. The validity of Cox models was verified by testing the proportional hazard assumption with the Grambsch and Ternau score test [19], the goodness of fit on 4 quantiles of risk with the May and Hosmer test [20], and the global significance of the models with the likelihood ratio (LR) test. All calculations were performed with version 14.2 of the Stata statistical software [21].

The optimal threshold values of PLR and NLR were determined as follows: the effect of individual PLR and NLR thresholds was studied as independent variables in logistic regression models designed to assess 5-year survival. Rounding PLR and NLR values to the accuracy of 1 and of 0.1, we calculated the area under the ROC curve (AUROC) for every threshold level. The optimal threshold values most effective for distinguishing among the patients by 5-year mortality, as determined based on AUROC peaks were 107 for PLR (Figure 1A) and 3.9 for NLR. (Figure 1B)

## 3. Results

The mean age of the subjects was 58.1 years (± SD: 8.7); and the proportion of females was 19.9%. The median duration of follow-up was 36 months, whereas the shortest duration was 4.9 and the longest was 161 months. Sixty-three patients died during the study, and 43 of them from malignant disease. Recurrence occurred in 72 patients. Median survival was 124.4 months (95% CI 71.6–∞), whereas median disease-free survival was 78.5 months (95% CI 33.5–∞).

The distributions of both PLR and NLR were skewed to the right (Figure 2). Median PLR was 113 (IQR: 84.5 to 160), and its value reached or exceeded the threshold of 107 in 53.9% of the patients. Median NLR was 2.8 (IQR: 2.1 to 3.8), and 20.5% of the patients had an NLR value corresponding to the threshold of 3.9 or higher—the latter was observed only in combination with PLR values higher than the threshold. PLR was elevated in 33.3% of the patients, and neither PLR nor NLR was elevated in 46.2%. Considering the demographical parameters and the characteristics of the malignancy, a significant relationship of PLR and NLR values was detected only with tumour stage (Kruskal–Wallis test *p* < 0.001, and *p* = 0.004), and with the primary extension of the tumour (Figure 3). Red blood cell count (median: 4.57; IQR: 4.31 to 4.93) was not correlated with PLR (*r* = −0.019, *p* = 0.811) or with NLR (*r* = −0.023, *p* = 0.770). Platelet count (median: 270 G/L; IQR: 220.5 to 333) exhibited moderate correlation with PLR (*r* = 0.680, *p* < 0.001), and weak correlation with NLR values (*r* = 0.227, *p* = 0.004). The study population was divided into three groups by primary treatment modality—that is, into groups formed by patients treated with surgery, by radio-/chemotherapy, or radiotherapy. Considering primary and secondary therapies together, 136 patients underwent surgery, 65 were treated with chemotherapy, and 49 with radiotherapy.

The subjects were also studied after grouping by the site of origin of the tumour (Table 1). The age distribution, tumour stage, the proportion of patients with a PLR value higher than the threshold, and of those surviving for 5 years differed significantly by tumour location. The proportion of patients with PLR values higher than the threshold was the largest (71.7%) among those with tumours of the hypopharynx, and the smallest (25.0%) among those with oropharyngeal malignancies.

Over a period of five years, 24.6% of the patients with a PLR value under the threshold of 107 died, whereas the mortality rate was 46.4% among those with a PLR ≥107—the difference between these two groups was statistically significant (log-rank test χ^2^
_(df = 1)_ = 8.14, *p* = 0.004). Over five years, tumour recurrence occurred in 37.1% of patients with a PLR value lower than the threshold, and in 52.2% of those with a PLR value higher than the threshold. The difference between the two groups was significant also with regard to disease-free survival (log-rank test χ^2^
_(df = 1)_ = 4.03, *p* = 0.045), as well as tumour-specific mortality (log-rank test χ^2^
_(df = 1)_ = 6.89, *p* = 0.009).

The mortality rate of patients with an NLR value under the threshold of 3.9 was 32.3%, whereas 56.5% of patients with NLR ≥ 3.9 died. The five-year recurrence rate was 43.0% in the former and 55.2% in the latter group; this difference was not statistically significant (log-rank test χ^2^
_(df = 1)_ = 2.21, *p* = 0.138). By contrast, a statistically significant difference in tumour-specific survival was noted between the two groups (log-rank test χ^2^
_(df = 1)_ = 7.43, *p* = 0.006).

We also studied the correlation between the combination of PLR and NLR values and five-year mortality. The subjects could be categorised into the following three groups: (1) Both NLR and PLR were higher than the threshold, (2) only PLR was higher than the threshold, and (3) both NLR and PLR were lower than the threshold. We found significant differences among the three groups in overall survival (log-rank test χ^2^
_(df = 2)_ = 12.78, *p* = 0.002), and in tumour-specific survival (log-rank test χ^2^
_(df = 2)_ = 9.92, *p* = 0.007). However, disease-free survival was not significantly different among these three groups (log-rank test χ^2^
_(df = 2)_ = 4.45, *p* = 0.108). The chance for five-year survival was the highest in patients who had both NLR and PLR values under the threshold, and it was the lowest in those in whom both ratios were elevated (Figure 4).

We analysed the five-year mortality, tumour-specific mortality, and disease-free survival with multivariate Cox-regression, using the impact of PLR and of NLR, as well as the joint effect of these two markers as independent variables. Age, gender, tumour stage, the grade of differentiation, initial region, red blood cell count, and thrombocytosis were used as control variables. In our study population, NLR values higher than the threshold were observed only in combination with elevated PLR values and therefore, studying the effect of isolated NLR elevation was not possible. The assumption for the proportional hazard was not fulfilled in the models designed to study tumour-specific mortality and accordingly, their coefficients are of limited validity. Among the control variables, male gender and advanced tumour stage were associated with an enhanced risk with respect to survival. Further, the likelihood of tumour recurrence was significantly correlated with male gender and with the grade of tumour differentiation. There was an inverse relationship between red blood cell count and the risk of mortality and of recurrence—that is, the elevation of red blood cell count was associated with a lower risk of death or tumour recurrence (Table 2). In models controlled for the extension of T stage—analysed instead of or in combination with TNM stage—the conclusions regarding overall survival remained unchanged. The models containing platelet count as a continuous control variable similarly did not influence the negative prognostic role of NLR for overall survival. However, the coefficients of the model of tumour-specific mortality did not prove stable upon adjusting the control variables, and the NLR value lost significance following the addition of T stage as a control variable.

## 4. Discussion

Patients with a high platelet count are at an increased risk of cancer, as well as certain malignancies which exhibit a closer relationship with thrombocytosis [22,23]. Previous studies confirmed the prognostic significance of platelet count in a variety of solid (renal-cell, hepatocellular, lung, colorectal, gastric, and gynaecological) tumours [24,25,26,27,28,29].

Similar to elevated platelet count, high PLR has been found to be associated with worse survival in a variety of solid (hepatocellular, ovarian, cervix, colorectal, bile duct, breast, lung, renal cell, gastric, prostate, and oesophageal) tumours [30,31,32,33,34,35,36,37,38,39,40]. Moreover, several studies confirmed the relationship between elevated NLR and a worse prognosis in patients with malignant oesophageal [41], colorectal [42,43], pancreas [44,45], prostate [46,47], hepatic [48,49], bile duct [50], breast [51], gastric [52,53], cutaneous melanocytic [54], thyroid medullary [55], and lung [56] tumours. It is important to note, however, that NLR may be elevated also in many benign disorders [57,58].

The NLR and PLR cut-off levels chosen in various studies varied in a wide range (NLR: 2.0 to 4.2 [55,59,60,61,62,63], PLR: 105.3–170 [55,62,63]). In our study, the optimal threshold value, determined with the statistical method detailed herein, was 107 for PLR and 3.9 for NLR.

The association between elevated PLR or NLR and worse prognosis was observed also in HNSCC [61,63,64,65,66], although these studies were limited to tumours of a specific location or stage, or to patients receiving a certain treatment modality. According to our findings, elevated PLR was associated with a significantly inferior five-year survival and disease-free survival, as well as with greater tumour-specific mortality. In case of elevated NLR, by contrast, only the increase of tumour-specific mortality was significant. Primary treatment modality had no significant influence on NLR/PLR values.

RASSOULI et al. found that when considered together as a prognostic marker for survival, the combination of NLR and PLR proves as reliable as does TNM staging [62]. Our study yielded similar findings, because the chance for five-year survival was the highest if neither NLR nor PLR exceeded the threshold level. Conversely, survival rate was the lowest when both NLR and PLR were elevated.

In another study, the prognostic value of NLR was also assessed by the primary location of naso-/oro-/hypopharyngeal or laryngeal tumours; however, only patients undergoing radiotherapy were included. A significant relationship was demonstrated for nasal, hypopharyngeal, and laryngeal— but not for oropharyngeal—tumours [60]. In our study, we did not detect any significant differences between NLR and the different primary locations of the tumours. By contrast, the proportion of patients with PLR values higher than the threshold differed significantly by primary site: it was the highest with hypopharyngeal and the lowest with oropharyngeal tumours. HPV status was not available for our analysis, but the frequent occurence of HPV-positive tumours in the oropharynx may give an explanation for the above-mentioned lower incidence of high PLR value at this tumour site. Some studies suggest that HPV infection might alter inflammatory response by altering white blood cell distribution and thus affect the PLR and NLR values [67]. Rosculet et al. obtained similar results in their retrospective study. They found that NLR is an indicator of both disease-free and overall survival, but it does not have independent prognostic significance when HPV status is incorporated into multivariate analysis [68].

When used as prognostic factors, the advantage of NLR and PLR is that both can be determined by routine laboratory methods. However, both are influenced by inflammation or anaemia, and this may undermine their reliability. This is why we defined active inflammation among the exclusion criteria, as well as used red blood cell count as a control variable during the multivariate analysis. According to the multivariate Cox regression analysis controlled in this fashion, PLR had no significant influence on mortality risk in any of the models. By contrast, NLR values higher than the threshold represented an enhanced risk both for overall and for tumour-specific mortality. Neither PLR nor NLR had a significant impact on the risk of tumour recurrence.

## 5. Conclusions

NLR values over the threshold of 3.9 (determined using the data of our patients) may be regarded as an independent risk factor for five-year survival in malignancies of the head and neck region. NLR was closely associated, both with the size of the primary tumour and with the stage of the malignant disease. Furthermore, NLR values higher than the threshold represented an enhanced risk regarding overall survival, as well as in other models controlled for the extension of the primary tumour, for the stage of the malignant disease, and for other demographical parameters, or disease characteristics. In view of the foregoing, therefore, we recommend determining NLR in routine clinical practice as a risk factor for five-year overall survival. On the other hand, we could not demonstrate a significant relationship between tumour-free survival and NLR, and this relationship was not consistent with regard to tumour-specific survival.

Nevertheless, it would be desirable to undertake a similar study in a larger population of patients, and with other malignancies (of the thyroid or salivary glands, for example) of the head and neck region.

## Figures and Tables

**Figure 1 ijerph-17-01742-f001:**
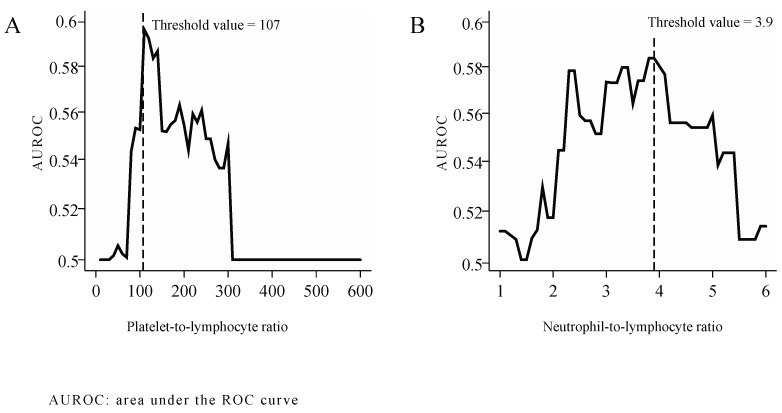
Threshold values of PLR (**A**) and NLR (**B**).

**Figure 2 ijerph-17-01742-f002:**
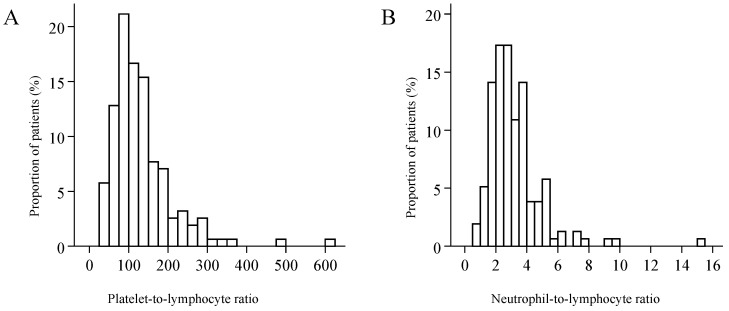
The distribution of the platelet-to-lymphocyte ratio (**A**), and of the neutrophil-to-lymphocyte ratio (**B**).

**Figure 3 ijerph-17-01742-f003:**
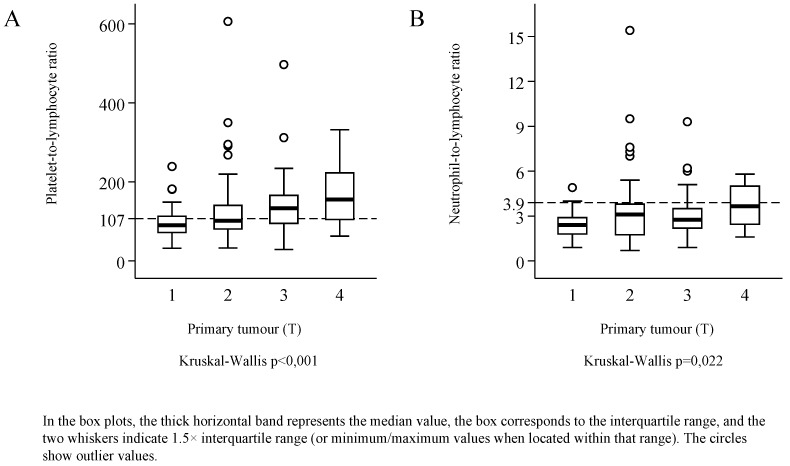
The platelet-to-lymphocyte ratio (**A**), and the neutrophil-to-lymphocyte ratio (**B**) in subgroups defined according to the stage and the location of the primary tumour (T).

**Figure 4 ijerph-17-01742-f004:**
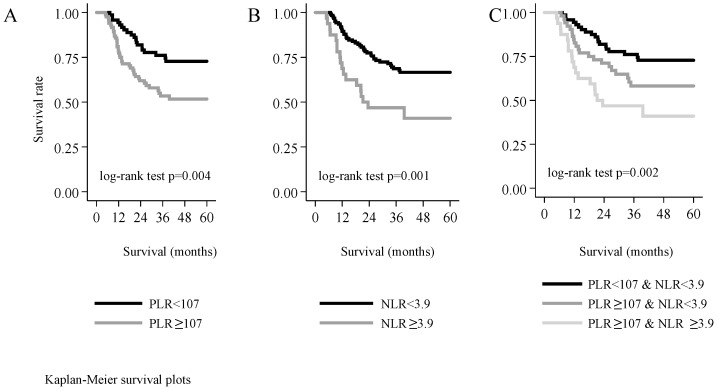
Five-year survival as a function of the platelet-to-lymphocyte ratio (**A**), of the neutrophil-to-lymphocyte ratio (**B**), or of both markers (**C**).

**Table 1 ijerph-17-01742-t001:** Main patient characteristics by tumour location.

		All Patients	Hypo-Pharynx	Oropharynx	Oral Cavity	Sub-/Supraglottis	Glottis	Multiple Regions	
		N (%)	N (%)	N (%)	N (%)	N (%)	N (%)	N (%)	χ^2^
Gender	Female	31 (19.9)	8 (15.1)	7 (43.8)	1 (20.0)	7 (20.6)	7 (15.6)	1 (33.3)	*p* = 0.195
	Male	125 (80.1)	45 (84.9)	9 (56.2)	4 (80.0)	27 (79.4)	38 (84.4)	2 (66.7)	
Age	<65 years	116 (74.4)	49 (92.5)	13 (81.2)	4 (80.0)	23 (67.6)	25 (55.6)	2 (66.7)	*p* = 0.002
	≥65 years	40 (25.6)	4 (7.5)	3 (18.8)	1 (20.0)	11 (32.4)	20 (44.4)	1 (33.3)	
T stage	1	38 (24.4)	6 (11.3)	8 (50.0)	0 (0.0)	5 (14.7)	18 (40)	1 (33.3)	*p <* 0.001
	2	40 (25.6)	9 (17)	4 (25.0)	3 (60.0)	16 (47.1)	8 (17.8)	0 (0.0)	
	3	50 (32.1)	31 (58.5)	4 (25.0)	1 (20.0)	6 (17.6)	7 (15.6)	1 (33.3)	
	4	28 (17.9)	7 (13.2)	0 (0.0)	1 (20.0)	7 (20.6)	12 (26.7)	1 (33.3)	
N stage	0	82 (52.6)	14 (26.4)	5 (31.2)	4 (80.0)	18 (52.9)	40 (88.9)	1 (33.3)	*p <* 0.001
	1	30 (19.2)	15 (28.3)	7 (43.8)	0 (0.0)	6 (17.6)	1 (2.2)	1 (33.3)	
	2	40 (25.6)	21 (39.6)	3 (18.8)	1 (20.0)	10 (29.4)	4 (8.9)	1 (33.3)	
	3	4 (2.6)	3 (5.7)	1 (6.2)	0 (0.0)	0 (0)	0 (0.0)	0 (0.0)	
TNM stage	1	26 (16.7)	2 (3.8)	3 (18.8)	0 (0.0)	3 (8.8)	18 (40)	0 (0.0)	*p <* 0.001
	2	24 (15.4)	4 (7.5)	1 (6.2)	2 (40.0)	9 (26.5)	8 (17.8)	0 (0.0)	
	3	44 (28.2)	20 (37.7)	8 (50)	1 (20.0)	7 (20.6)	7 (15.6)	1 (33.3)	
	4	62 (39.7)	27 (50.9)	4 (25)	2 (40.0)	15 (44.1)	12 (26.7)	2 (66.7)	
Grade of differentiation	1	20 (12.8)	4 (7.5)	0 (0.0)	2 (40.0)	3 (8.8)	11 (24.4)	0 (0.0)	*p* = 0.270
	2	54 (34.6)	17 (32.1)	6 (37.5)	1 (20.0)	15 (44.1)	14 (31.1)	1 (33.3)	
	3	37 (23.7)	11 (20.8)	4 (25)	1 (20.0)	9 (26.5)	11 (24.4)	1 (33.3)	
	4	2 (1.3)	0 (0.0)	1 (6.2)	0 (0.0)	0 (0.0)	1 (2.2)	0 (0.0)	
	Unknown	43 (27.6)	21 (39.6)	5 (31.2)	1 (20.0)	7 (20.6)	8 (17.8)	1 (33.3)	
Platelets-to-lymphocytes ratio	<107	72 (46.2)	15 (28.3)	12 (75.0)	2 (40.0)	16 (47.1)	26 (57.8)	1 (33.3)	*p* = 0.011
	≥107	84 (53.8)	38 (71.7)	4 (25.0)	3 (60.0)	18 (52.9)	19 (42.4)	2 (66.7)	
Neutrophils-to-lymphocytes ratio	<3,9	125 (80.1)	39 (73.6)	15 (93.8)	4 (80.0)	26 (76.5)	38 (84.4)	3 (100.0)	*p* = 0.435
	≥3,9	31 (19.9)	14 (26.4)	1 (6.2)	1 (20.0)	8 (23.5)	7 (15.6)	0 (0.0)	
5-year mortality	Deceased	58 (37.2)	28 (52.8)	4 (25.0)	1 (20.0)	16 (47.1)	8 (17.8)	1 (33.3)	*p* = 0.01 *
Total		156	53	16	5	34	45	3	

* log-rank test.

**Table 2 ijerph-17-01742-t002:** Multivariate Cox regression analysis of five-year and of tumour-specific mortality, and of tumour recurrence.

		Overall Mortality			Tumour-Specific Mortality			Tumour Recurrence		
PLR	≥107	1.499			1.499			1.651		
NLR	≥3.9		2.122 *			2.122 *			1.800	
PLR and NLR aggregately	PLR ≥ 107 and NLR < 3.9			1.153			1.153			1.435
	PLR ≥ 107 and NLR ≥ 3.9			2.291 *			2.291 *			2.180 *
Age ^a^	≥65 years	0.906	1.045	1.018	0.906	1.045	1.018	0.825	0.876	0.860
Gender ^b^	Male	2.460	2.730 *	2.732 *	2.460	2.730 *	2.732 *	3.530 **	3.809 **	3.840 **
Stage ^c^	2	6.690 ^g^	6.802 ^g^	6.616	6.690 ^g^	6.802 ^g^	6.616	2.198	2.269	2.175
	3	7.315 ^g^	7.698 ^g^	7.439	7.315 ^g^	7.698 ^g^	7.439	1.293	1.443	1.306
	4	14.44 *^g^	13.40 *^g^	12.87 *	14.44 *^g^	13.40 *^g^	12.87 *	2.591	2.733 *	2.447
Grade of differentiation ^d^	2	1.778	1.911	1.941	1.778	1.911	1.941	4.274 *	4.518 *	4.741 *
	3	1.341	1.534	1.540	1.341	1.534	1.540	2.539	2.762	2.854
	4	-	-	-	-	-	-	4.159	4.020	4.845
	Unknown	1.954	2.210	2.237	1.954	2.210	2.237	2.977	3.272	3.393
Location ^e^	Oropharynx	0.817	0.796	0.833	0.817	0.796	0.833	1.285	1.196	1.331
	Oral cavity	0.546	0.549	0.559	0.546	0.549	0.559	12.29 **	13.21 **	13.30 **
	Sub-/supraglottis	1.221	1.152	1.189	1.221	1.152	1.189	1.330	1.160	1.264
	Glottis	0.631	0.560	0.577	0.631	0.560	0.577	1.100	0.991	1.064
	Multiple locations	0.475	0.595	0.593	0.475	0.595	0.593	0.633	0.618	0.676
RBC		0.503 *	0.508 **	0.509 **	0.503 *	0.508 **	0.509 **	0.508 **	0.512 **	0.510 **
Platelet count ^f^	≥400 G/L	0.906	1.045	1.018	0.906	1.045	1.018	0.825	0.876	0.860
Number of patients		156	156	156	156	156	156	148	148	148

The table shows hazard ratios; values over 1 indicate an enhanced risk. ^a^ Base: <65 years, ^b^ Base: Female, ^c^ Base: Stage 1, ^d^ Base: Grade 1, ^e^ Base: Hypopharynx, ^f^ Base: <400 G/L, ^g^ Wald-test *p* < 0.05, * *p* < 0.05, ** *p* < 0.01

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
