# Peer review of "Neutrophil-To-Lymphocyte and Platelet-To-Lymphocyte Ratios as Prognostic Markers of Survival in Patients with Head and Neck Tumours—Results of a Retrospective Multicentric Study"

_ijerph, 2020, doi:10.3390/ijerph17051742_

Round 1
Reviewer 1 Report
In their manuscript the authors retrospectively analyze the effect of NLR and PLR on the survival of HNSCC patients treated with surgery or (chemo)radiation. It is a strength of this study to evaluate the effect of NLR/PLR in a large cohort of HNSCC employing sound statistical methodology. On the other hand, the novelty of this analysis is limited, since it is has been published previously that NLR/PLR are prognostic factors in HNSCC Some major questions arose, when reviewing the manuscript and major issues should be addressed: - The introduction section should be streamlined and lay a focus on NLR/PLR in HNSCC. It is unclear why pro-inflammatory cytokines (line 61-65) are outlined, while not measured in this study. - The method section should be revised: Why are the ROC curves not shown in the results section? Determining an optimal threshold for NLR/PLR is one of the key findings of the study and should be shown. - On the other hand line 135-139 is not a result, but this should be stated in the methods section. - Which TNM version of the staging system was employed? -The discussion section should put the findings into the context of the current literature and is too general. Apart from that, recent studies are not cited and/or discussed such as Rosculet et al.; Head and Neck 2017. This study did not show prognostic significance in HPV positive OPC, which is in line with Cho et al. for OPC in general. How do the authors explain their findings in the light of those 2 studies? It should be stated in the discussion section as well that HPV status was not available for this analysis. - What threshold levels were given in the literature for HNSCC? Different findings should be discussed as well.
Author Response
Response to Reviewer 1 Comments
Point 1: The introduction section should be streamlined and lay a focus on NLR/PLR in HNSCC. It is unclear why pro-inflammatory cytokines (line 61-65) are outlined, while not measured in this study.
Response 1: I streamlined the introduction section, I only mention paraneoplastic thrombocytosis and focus on NLR/PLR in HNSCC. I agree, pro-inflammatory cytokines do not belong closely to the subject, I deleted this part. (deletion at line 46 and 47)
Point 2: The method section should be revised: Why are the ROC curves not shown in the results section? Determining an optimal threshold for NLR/PLR is one of the key findings of the study and should be shown.
Response 2: I inserted a figure with the ROC curves. (line 154)
Point 3: Line 135-139 is not a result, but this should be stated in the methods section.
Response 3: I moved the above-mentioned part from the results to the methods section. (move to line 128)
Point 4: Which TNM version of the staging system was employed?
Response 4: The 7th version of TNM classification was employed, I mentioned it at the methods section. (line 129)
Point 5: The discussion section should put the findings into the context of the current literature and is too general. Apart from that, recent studies are not cited and/or discussed such as Rosculet et al.; Head and Neck 2017. This study did not show prognostic significance in HPV positive OPC, which is in line with Cho et al. for OPC in general. How do the authors explain their findings in the light of those 2 studies? It should be stated in the discussion section as well that HPV status was not available for this analysis.
Response 5: I mentioned the study of Rosculet et al and confront their, Cho's and our findings. I stated at the discussion section that HPV status was not available for our analysis. (line 289-296)
Point 6: What threshold levels were given in the literature for HNSCC? Different findings should be discussed as well.
Response 6: The range of the threshold levels in the literature are given in the discussion section. (line 267-268)
Reviewer 2 Report
The manuscript describes the usefulness of NLR (and PLR) as predictors of survival in HNSCC patients. It is interesting concept to utilize the combination of NLR and PLR.
P.5 L.99-100: “Platelet, neutrophil, and lymphocyte counts measured at the earliest month before treatment.” Does that mean a blood test at the first visit or preoperative examination? I am wondering whether the value of NLR or PLR could be evaluated by only a single blood sampling, because values of blood tests including blood cell count are seemingly variable under normal conditions from time to time. What was the basis of this measurement design?
P7.L145: “The distributions of both PLR and NLR were skewed.” I do not know a lot about these biomarkers and I would appreciate if you could provide information about normal ranges or distributions of these measurements.
P.9 L.170-173: The proportion of patients with PLR values higher than the threshold was the largest (71.7%) among those with tumors of the hypopharynx, and the smallest (25.0%) among those with oropharyngeal malignancies. What makes these difference of PLR values according to the site of the origin of the tumor?
P.22 L.285-286: “We recommend determining NLR in routine clinical practice as a risk factor for 5-year overall survival.” I desire better understanding of the great advantage of this biomarker. How do you think about the primary use case in clinical practice?
Author Response
Response to Reviewer 2 Comments
Point 1: “Platelet, neutrophil, and lymphocyte counts measured at the earliest month before treatment.” Does that mean a blood test at the first visit or preoperative examination? I am wondering whether the value of NLR or PLR could be evaluated by only a single blood sampling, because values of blood tests including blood cell count are seemingly variable under normal conditions from time to time. What was the basis of this measurement design?
Response 1: These are the results of the pretreatment examination. If there were more blood samples taken, we took into consideration the one that is closest in time to the determination of disease stage. Certain conditions, such as synchronous tumour, inflammatory diseases, thromboembolic events, corticosteroid therapy and treatment with platelet aggregation inhibitors might cause changes in the blood cell count, these conditions appear among the exclusion criteria.
Point 2: “The distributions of both PLR and NLR were skewed.” I do not know a lot about these biomarkers and I would appreciate if you could provide information about normal ranges or distributions of these measurements.
Response 2: A recent study based on a sample of 413 active subjects of good health showed that the normal NLR values are between 0.78 and 3.53, the mean was 1.65 (Forget et al, BMC Res Notes, 2017). An other study providing reference values for NLR and PLR is based on a cohort of more than 10.000 patients. According to these data, the mean NLR was also 1.65, and the mean PLR was 132.4. (Lee et al, Medicine, Baltimore, 2018).
Point 3: The proportion of patients with PLR values higher than the threshold was the largest (71.7%) among those with tumors of the hypopharynx, and the smallest (25.0%) among those with oropharyngeal malignancies. What makes these difference of PLR values according to the site of the origin of the tumor?
Response 3: HPV-positive tumors of the oropharynx are frequent, and some studies suggest that HPV infection might alter inflammatory response by altering white blood cell distribution and thus affecting the PLR and NLR values. I mentioned this in the revised manuscript, at the discussion section. (line 189-293)
Point 4: “We recommend determining NLR in routine clinical practice as a risk factor for 5-year overall survival.” I desire better understanding of the great advantage of this biomarker. How do you think about the primary use case in clinical practice?
Response 4: Patients with worse survival might need more aggressive, multimodal treatment and closer follow-up. If you can predict survival before treatment more precisely with a simple, routine method, it could enhance survival results.
Round 2
Reviewer 1 Report
The authors satisfactorily addressed all issues raised